# Loci Controlling Adaptation to Heat Stress Occurring at the Reproductive Stage in Durum Wheat

**Khaoula El Hassouni** [1,2]**, Bouchra Belkadi** [2]**, Abdelkarim Filali-Maltouf** [2]**,**
**Amadou Tidiane-Sall** [1,3]**, Ayed Al-Abdallat** [4]**, Miloudi Nachit** [1,2] **and Filippo M. Bassi** [1,*]

[1] International Center for the Agricultural Research in Dry Area (ICARDA), Rabat 10000, Morocco
[2] Faculty of sciences, University of Mohammed V, Rabat 10000, Morocco
[3] Institut Sénégalais de Recherches Agricoles (ISRA), Saint-Louis 46024, Senegal
[4] Faculty of Agriculture, The University of Jordan, Amman 11942, Jordan
[*] Correspondence: f.bassi@cgiar.org; Tel.: +212614402717

**Abstract:** Heat stress occurring during the reproductive stage of wheat has a detrimental effect on productivity. A durum wheat core set was exposed to simulated terminal heat stress by applying plastic tunnels at the time of flowering over two seasons. Mean grain yield was reduced by 54% compared to control conditions, and grain number was the most critical trait for tolerance to this stress. The combined use of tolerance indices and grain yield identified five top performing elite lines: Kunmiki, Berghouata1, Margherita2, IDON37-141, and Ourgh. The core set was also subjected to genome wide association study using 7652 polymorphic single nucleotide polymorphism (SNPs) markers. The most significant genomic regions were identified in association with spike fertility and tolerance indices on chromosomes 1A, 5B, and 6B. Haplotype analysis on a set of 208 elite lines confirmed that lines that carried the positive allele at all three quantitative trait loci (QTLs) had a yield advantage of 8% when field tested under daily temperatures above 31° C. Three of the QTLs were successfully validated into Kompetitive Allele Specific PCR (KASP) markers and explained >10% of the phenotypic variation for an independent elite germplasm set. These genomic regions can now be readily deployed via breeding to improve resilience to climate change and increase productivity in heat-stressed areas.

**Keywords:** heat stress; durum wheat; yield; tolerance; fertility; climate change; resilience

## 1. Introduction

Heat stress is a major environmental constraint to crop production. Terminal heat stress is defined as a rise in temperature that occurs between heading and maturity. When this stress matches with the reproductive phase of the wheat plant, it affects anthesis and grain filling, resulting in a severe reduction in yield [1]. High temperatures at the time of flowering cause floret sterility via pollen dehiscence [2], decrease photosynthetic capacity by drying the green tissues, and reduce starch biosynthesis [1,3]. These in turn result in a negative effect on grain number and weight [4–7]. The optimum growing temperature for wheat during pollination and grain filling phases is 21 °C [8,9], and for each increase of 1 °C above it is estimated a decline of 4.1% to 6.4% in yield [10]. Environmental temperatures have been increasing over the last century and more frequent heat waves are predicted in the next decades [11–13]. Therefore, breeding for tolerance to chronic as well as short term heat stress is a major objective worldwide [14–19]. Breeding selection would benefit by a better understanding of traits associated with tolerance to high temperatures, as well as the identification of the genomic regions controlling these traits.

In wheat, a large number of quantitative trait loci (QTLs) has been identified under heat stress via linkage analysis and genome-wide association study (GWAS) for yield, yield related traits, and some physiological traits such as chlorophyll content, chlorophyll fluorescence, and canopy temperature [20–27]. Grain number per spike and chlorophyll content were found to be the most critical traits for adaptation to warm conditions [24,25,28]. Heat stress reduces leaf chlorophyll content [29] affecting the amount of carbohydrates transported to the grains and final grain weight and size. High temperatures around anthesis reduce the number of grains per spike due to a decrease in spike growth and development, and an increase in ovules abortion [2,25,29,30]. To the best of our knowledge, molecular markers associated with heat tolerance are not generally used in wheat breeding programs [31–33]. The limited understanding of genes underlying physiological mechanisms and the regulation of yield components in wheat, and the lack of cloned major QTL for traits associated with heat tolerance has restricted the improvement in breeding for tolerance to this stress.

In the current study, a set of durum wheat lines were heat stressed by imposing a > 10 °C raise in maximum daily temperatures via the deployment of plastic tunnels at the time of flowering. GWAS studies allowed the identification of major QTLs controlling the adaptation to this stress and these were validated for marker assisted selection (MAS) in an independent germplasm set for rapid deployment via breeding.

## 2. Materials and Methods

### 2.1. Plant Material

A subset of 42 durum wheat inbred lines were selected from a global collection of 384 genotypes based on their similarity in flowering time and identified genetic diversity [34]. Briefly, the complete collection is highly diverse and includes 96 durum wheat landraces from 24 countries, and 288 modern lines from nine countries and two International research centers CIMMYT and ICARDA. The subset selected for this study includes 34 ICARDA and CIMMYT lines, five cultivars and one landrace. The list of the 42 genotypes and their details are provided in Table S1.

A second subset of 208 modern entries was also obtained from the global collection and field tested under severe high temperatures during 2014–2015 and 2015–2016 seasons along the Senegal River in Kaedi, Mauritania. Full details on this field experiment have been published in Sall et al. [35].

The third and final set was used for Kompetitive Allele Specific PCR (KASP) markers validation and it was composed of 94 ICARDA's elite lines that constituted the 2017 international nurseries 40th International Durum Yield Trial (IDYT) and 40th International Durum Observation Nurseries (IDON). This set was also tested at the station of Kaedi along the Senegal River in season 2015–2016.

### 2.2. Field Experiment Conditions and Phenotyping

The first subset of 42 entries was grown at Marchouch station (33°34′3.1″ N, 6°38′0.1″ W) in Morocco during two successive crop seasons (2015–2016 and 2016–2017). Each entry was sown in mid-November on a plot surface of 1.5 $m^2$ per genotype at a sowing density of 300 plants per $m^2$. The experiment was an alpha lattice with two replications, block size of six, and two treatments arranged in split-plot. Each six genotypes were arranged in close proximity to maximize competition between the genotypes, and compose one block of 9 $m^2$. Each block was surrounded by a border of barley to avoid border effect. Each block was spaced 1 m apart to allow the application of the plastic tunnel. The two treatments were normal rainfed conditions and plastic tunnel-mediated heat stress. The normal treatment followed standard agronomic practices with a base pre-sowing application of 50 Kg $ha^{-1}$ of N, P, and K. At stage 15 of Zadok's (Z) scale herbicide was applied in a tank mixture (Pallas + Mustang at 0.5 L $ha^{-1}$) to provide protection against both monocots and dicots. At Z17 ammonium nitrate was provided to add 36 kg $ha^{-1}$ of N and a final application of urea was used to add 44 kg $ha^{-1}$ of N before booting (Z39). Weeds were also controlled mechanically to ensure clean plots. The soil of the experimental station is clay-vertisol type. The available on season moisture was 234 and 280 mm

for 2015–2016 and 2016–2017, respectively, during the growing season, whereas the average daily temperature was 14.1 °C for the first year and 13.5 °C for the second year. The heat-stress treatment followed the same agronomic practices, with the difference that at the time of booting (Z45) a 10 m$^2$ and 1.5 m high plastic tunnel was placed over each block (Figure 1) and left there until early dough stage (Z83). An electronic thermometer (temperature data logger) was placed in the middle of each block (normal and heat stressed) to reveal that the temperatures were up to 16° C higher inside the plastic tunnels, to reach a maximum of 49 °C (Figure 1). Marchouch is a drought prone site, and no rainfall occurred after Z45 in any of the two field seasons.

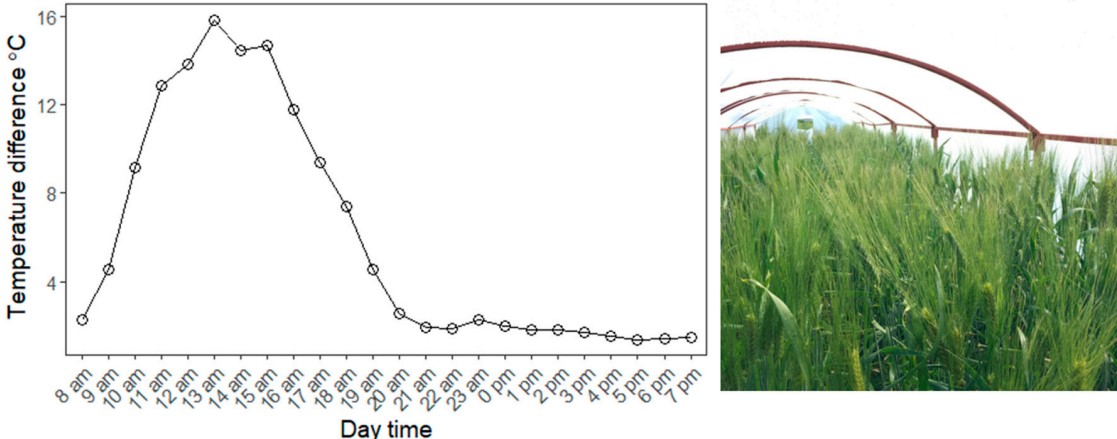

**Figure 1.** Mean temperature difference of 18 days over two seasons between the plastic tunnel-mediated heat stress and normal field conditions between 8 a.m. and 8 p.m., and a picture of the plastic tunnel at 9 a.m.

The following traits were recorded: days to heading (DTH) measured at the moment when the awns became visible, plant height (PH) measured from the ground to the top of the highest spike excluding the awns, and the number of fertile spikes per meter square (Spkm$^2$) was counted in a 0.25 m$^2$ area. The whole plot was harvested by hand and the dry biomass (Biom) was weighed before threshing. Grain yield (GY) was weighed for each plot and expressed as kg ha$^{-1}$. The weight of a thousand kernels (TKW) was expressed in grams. The harvest index (HI) was calculated as the ratio between GY and Biom. The grain number per spike (GNSpk) was derived from dividing grain number per meter square by Spkm$^2$ as follows:

$$\text{Grain number/m}^2 = \frac{\text{Grain weight of the plot}}{1.5\text{m}^2 \times \frac{\text{TKW}}{1000}} \tag{1}$$

$$\text{GNSpk} = \frac{\text{Grain number/m}^2}{\text{Spkm}^2} \tag{2}$$

The second and third sets were field tested in Kaedi, Mauritania (16°14″ N; 13°46″ W) during season 2014–2015 and 2015–2016 where the temperature reached a maximum of 41 °C and an average maximum daily temperature of 34 °C throughout the season. The trial was carried out under augmented design with a plot surface of 4.5 m$^2$. Standard agronomic management practices were adopted. Full details for this experiment are published elsewhere [35].

*2.3. Data Analysis*

A mixed linear model was run using the lme4 package [36] in R [37] to obtain best linear unbiased estimates (BLUEs) of the normally distributed traits. For count traits (DTH, Spkm$^2$, GNSpk), the generalized mixed linear model was used to get the BLUEs by Proc GLIMMIX in SAS. In both models, genotype, treatment, year, and replication were considered as fixed effects and block as random effect

nested in treatment and year. Broad-sense heritability was calculated based on variance components from random model using the method suggested by DeLacy et al. [38]:

$$H^2 = \frac{\sigma^2 g}{\sigma^2 g + \frac{\sigma^2 GxT}{t} + \frac{\sigma^2 GxY}{y} + \frac{\sigma^2 GxTxY}{ty} + \frac{\sigma^2 e}{tyr}} \tag{3}$$

where: $\sigma^2_{G \times T}$ = genotype × treatment variance, $\sigma^2_{G \times Y}$ = genotype × year variance, $\sigma^2_{G \times Y \times T}$ = genotype × treatment × year variance, $\sigma^2_e$ = residual variance, $r$ is the number of replications per treatment, $t$ is the number of treatments, and $y$ is the number of years.

Box-and-whisker plots where constructed by ggplot2 package [39] using the BLUEs combined over year per each treatment. The relationship between the target trait GY and yield components (GNSpk, TKW, Biom, HI) was studied using the Pearson correlation coefficient and the additive regression model. The critical value of the correlation significance was determined at 0.30 for $p < 0.05$ and 0.39 for $p < 0.01$ for 40 df using the corrplot package [40]. The additive model incorporates flexible forms (i.e., splines) of the functions to account for non-linear relationship contrary to linear regression model estimated via ordinary least squares [41]. For the additive model, the effective degree of freedom term determines the nature of the relationship between the predictor and the response variables where EDF = 1 indicates linearity and EDF > 1 the non-linearity. The additive regression analysis was performed using the mgcv package [42].

Two stress tolerance indices were calculated to identify the heat tolerant genotypes. The stress susceptibility index (SSI) [43,44] was calculated as follows:

$$SSI = \frac{[1 - (Ys)/(Yp)]}{[1 - (\bar{Y}s)/(\bar{Y}p)]} \tag{4}$$

where Ys and Yp are yield values of the genotypes evaluated under heat stress and normal conditions, respectively, and $\bar{Y}s$ and $\bar{Y}p$ are the mean yields of the lines evaluated under heat stress and normal conditions, respectively.

The stress tolerance (TOL) [45] was calculated as follows:

$$TOL = Yp - Ys. \tag{5}$$

The classInt package [46] was used to identify the possible number of class intervals of the indices for the frequency distribution of the subset.

The cut-off value for tolerant vs. susceptible genotypes for SSI was equal to 1, with lines having SSI < 1 being stress tolerant. Regarding the TOL index, the smaller TOL values indicate the genotypes with low yield depression and hence more tolerant. The experiment-wide TOL mean (1608 kg ha$^{-1}$) was identified as the cut-off value for tolerant vs. susceptible. The emmeans package [47] based on ANOVA model was used to discriminate among the grain yield means of haplotypes.

*2.4. Genotyping and Marker-Trait Associations*

Details of the genotyping step of the core set and panel have been previously discussed in Kabbaj et al. [34] and Sall et al. [35]. Briefly, 7652 high-fidelity polymorphic single nucleotide polymorphism (SNPs) were obtained, showing less than 1% missing data, minor allele frequency (MAF) higher than 5%, and heterozygosity less than 5%. The sequences of these markers were aligned with a cut-off of 98% identity to the durum wheat reference genome [48] (available at: http://www.interomics.eu/durum-wheat-genome), to reveal their physical position. The average length of the Axiom probe is of 75 bp, hence the 2% allowed miss-match was set to account for the existence of 1 SNP within each sequence. A sub-set of 500 highly polymorphic SNPs were selected on the basis of even spread along the genome, and used to identify the existence of population sub-structure, which revealed the existence of 10 main sub-groups [34]. To avoid bias, these 500 markers were then removed

from all downstream association analysis. Linkage disequilibrium was calculated as squared allele frequency correlations ($r^2$) in TASSEL V 5.0 software [49], using the Mb position of the markers along the bread wheat reference genome. Linkage disequilibrium (LD) decay was estimated and plotted using the "Neanderthal" method [50]. The LD decay was measured at 51.3 Mb for $r^2 < 0.2$ as presented in Bassi et al. [51].

The genome wide association study (GWAS) was based on BLUEs of all the traits that displayed a significant treatment effect and the two stress tolerance indices. Two models were fitted and compared using two covariate parameters, Q (population structure) and K (Kinship). Q model was performed using a general linear model (GLM), and Q + K model using a mixed linear model (MLM). The best model for each trait was selected based on the quantile-quantile (Q-Q) plots [52]. Flowering time (DTH) was used as covariate in all analyses to remove the strong effects of flowering genes from the study. The value calculated for the LD decay of 51.3 Mb indicated that this association panel interrogated the 12,000 Mb of the durum wheat genome via 248 "loci hypothesis," and hence the Bonferroni correction for this panel was set to 3.1 LOD for $p < 0.05$ as suggested by Duggal et al. [53]. Local LD decay for $r^2 < 0.2$ was calculated for a 100 Mbp window around the marker with highest LOD for all marker-trait associations (MTAs) identified at a distance inferior to 104 Mbp (twice the LD decay). The MTAs that occurred at a distance inferior to twice the local LD were considered to belong to the same QTL. QTL associated to flowering time were removed from all downstream analyses (Table S2). A regression analysis was performed between the haplotype of the peak marker of each QTL to determine possible duplicate or homeolog loci. In addition, all the MTAs analyses were performed using Tassel 5 software [49].

### 2.5. Markers Conversion to KASP (Kompetitive Allele Specific PCR)

The array sequences of 20 markers associated to traits (MTA) were submitted to LGC Genomics for in-silico design of KASP primers using their proprietary software. Those that passed the in-silico criteria were purchased and used to genotype the independent validation set. For each marker that amplified and showed polymorphism, the regression cut-off between phenotype and haplotype was imposed at $r = 0.105$ following Pearson's critical value [54]. KASP markers AX-95260810, AX-94432276, and AX-95182463 were tested for association with grain yield, while AX-94408589 for association with biomass. In addition, the top 20 and worst 20 lines were considered as the true positive and true negative for heat tolerance. Hence, the accuracy was calculated as the ratio of the correct allelic call among all, sensitivity as the ratio of the correct positive allelic among the top 20 yielding lines, and specificity as the ratio of the correct negative (wt) allelic calls among the 20 worst yielding lines. The sequence of the validated KASP markers is provided in Table S3, or the primers can be ordered directly at LGC Genomics indicating the Axiom code used in this article.

## 3. Results

### 3.1. Agronomic Performance of the Genotypes and Sensitivity of Traits to Heat Stress

The combined analysis of variance across four environments (two different temperature treatments over two crop seasons) revealed significant genotypic differences for all traits measured (Table 1). The yield performance of the genotypes across environments averaged 2171 kg ha$^{-1}$ and ranged from 352 kg ha$^{-1}$ obtained under heat stress conditions for the lowest yielding line DWAyT-0215, to 4658 kg ha$^{-1}$ under normal conditions for the highest yielding line DWAyT-0217. The top yielding line under heat-stress was the ICARDA/Moroccan cultivar 'Faraj' with an average yield of 2249 kg ha$^{-1}$ over the two seasons.

**Table 1.** Descriptive statistics, component of trait variation, and heritability ($h^2$) among a set of 42 durum genotypes (G) tested under two treatments (T): normal and plastic tunnel-mediated heat stress during seasons 2015–2016 and 2016–2017.

| Trait | Acronym | Mean | Min | Max | Genetic Variance (%) | Treatment Variance (%) | G × T (%) | $h^2$ |
|---|---|---|---|---|---|---|---|---|
| Days to heading | DTH | 92 | 71 | 109 | 34 ** | 1$^{ns}$ | 1$^{ns}$ | 0.78 |
| Plant height (cm) | PH | 81 | 71 | 92 | 60 ** | 1$^{ns}$ | 16$^{ns}$ | 0.76 |
| Biomass (kg ha$^{-1}$) | Biom | 8407 | 4792 | 13,108 | 49 ** | 7 ** | 7 ** | 0.79 |
| Spikes number per m$^2$ | Spkm$^2$ | 524 | 370 | 640 | 14 ** | 1$^{ns}$ | 2 ** | 0.50 |
| Grain yield (kg ha$^{-1}$) | GY | 2171 | 352 | 4658 | 30 ** | 44 ** | 12 * | 0.63 |
| Harvest index (%) | HI | 26 | 1 | 50 | 15 ** | 34 ** | 13$^{ns}$ | 0.20 |
| Thousand kernel weight (g) | TKW | 36 | 27 | 45 | 48 ** | 1$^{ns}$ | 18 ** | 0.72 |
| Grain number per spike | GNSpk | 13 | 3 | 24 | 19 * | 29 ** | 16 ** | 0.46 |

*, ** Significant at the 0.05 and 0.01 probability levels, respectively.

The treatment effect was significant only for Biom, GY, HI, and GNSpk, whereas DTH, PH, Spkm$^2$, and TKW were not significantly affected by treatments (Figure 2). The yield components were all significantly reduced under heat stress except TKW that showed a slight increase for the genotypes exposed to heat. The genotypes tested under plastic-tunnels had 61%, 54%, 42%, and 17% lower average GNSpk, GY, HI and Biom, respectively, compared to control. Relatively high heritability was observed for all the phenological and agronomical traits except for HI that had the lowest heritability ($h^2$ = 0.20).

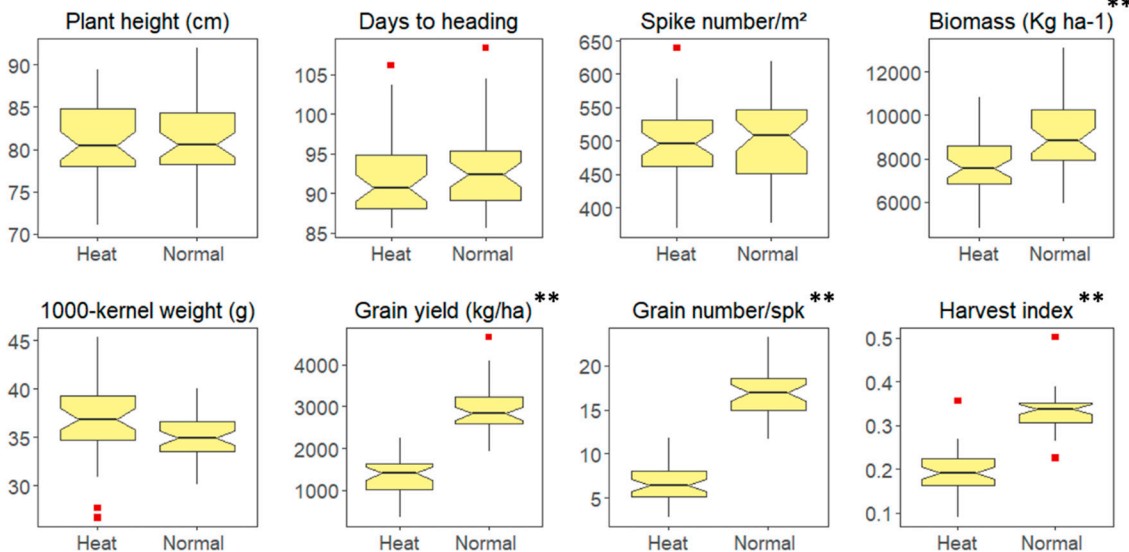

**Figure 2.** Boxplot of the best linear unbiased estimates (BLUEs) for various traits under two different environmental conditions (Heat: plastic tunnel-mediated heat stress and Normal) across two years. ** indicate significant difference between the means of control and heat-stressed plants at $p < 0.05$.

### 3.2. The Traits Interrelationship under Each Environmental Condition

Correlation analysis (Figure 3; Tables S4 and S5) was first conducted to investigate the interrelationship among all agronomic traits. Under both treatments, GNSpk had the highest association with GY ($r = 0.81$ under heat, $r = 0.67$ under normal), while Spkm$^2$ and TKW were the least correlated with GY. Biomass was also correlated with GY with $r = 0.61$ under heat and $r = 0.67$ under normal conditions. HI also showed a significant positive correlation with yield under both treatments, but its effect was stronger under heat stress ($r = 0.72$) than normal conditions ($r = 0.54$). DTH was not significantly correlated to any trait except HI ($r = -0.44$) under normal conditions.

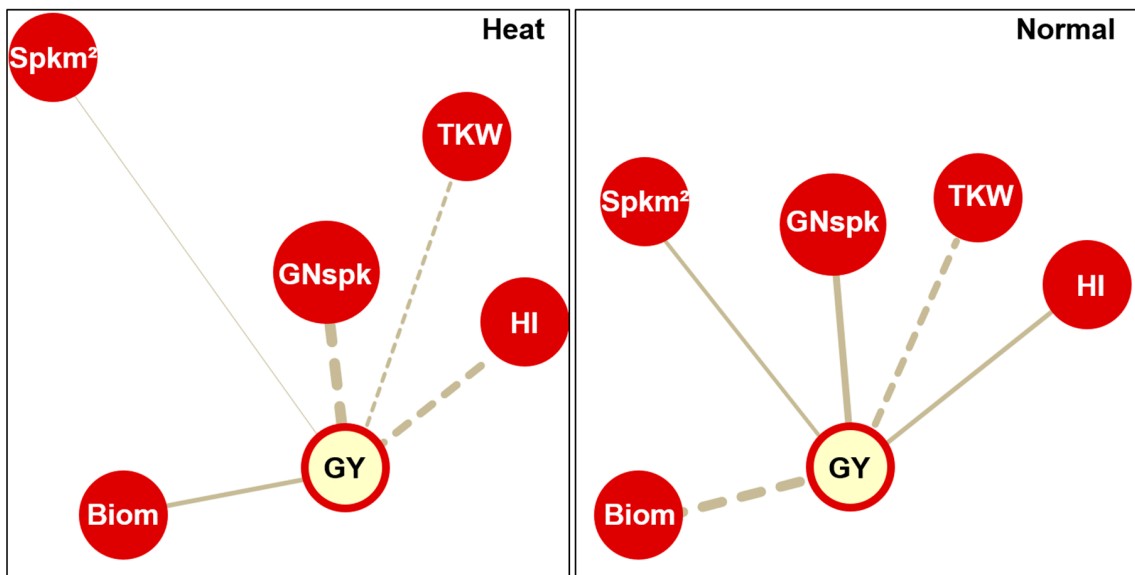

**Figure 3.** Relationships between grain yield (GY) and yield components (grain number per spike (GNspk), harvest index (HI), dry biomass (Biom), number of fertile spikes per meter square (Spkm$^2$), weight of a thousand kernels (TKW)) under plastic tunnel-mediated heat stress and normal conditions assessed by Pearson correlation and simple generalized additive model. The continuous grey line represents a linear relationship; the dashed grey line represents a non-linear relationship. The thickness of the line indicates the level of predictivity of the trait for GY. The length of the lines represents the correlation, the shorter the line the more the trait is correlated to GY.

Among yield components, the only significant and positive associations under the two environmental conditions were observed between Spkm$^2$, TKW, and Biom and between HI and GNSpk. Under heat conditions, a positive and significant correlation was noticed between GNSpk and Biom while under normal conditions HI was positively associated to TKW (Figure 3; Table S4).

The additive model was then used to further determine the nature of the relationship between GY and each predictor variable under normal and heat conditions (Figure 3; Table S5). The similarities observed between the two treatments in terms of the nature of relationship between GY and each of the predictors were the constantly linear and non-linear relationship between Spkm$^2$, TKW and the response variable GY, respectively.

GNSpk was considered the best predictor (deviance = 0.73%) with a complex relationship (EDF = 2.64) with GY under heat stress, whereas under normal conditions this trait was the second best predictor (deviance = 0.44%) with a linear relationship (EDF = 1). A similar trend was observed for HI in both treatments. Biom was found to be the best predictor (deviance = 0.52%) for GY with a non-linear relationship (EDF = 2.52) under normal conditions (Table S2; Figure S1).

### 3.3. Stress Tolerance Indices

Two different stress tolerance indices were calculated for GY: SSI and TOL (Figure 4). The genotypes showed wide variation for these indices. Seven SSI groups were identified with four having an SSI lower than 1 and the three remaining groups of genotypes having SSI > 1. The frequency distribution of the panel showed a wide variation and indicated the presence of susceptibility, with 45% of the genotypes falling in the very heat-susceptible class of SSI higher than 1, and only 7% of the lines showing high tolerance at SSI < 1. For TOL index, seven groups were also identified with 48% of the lines showing high yield depression and 5% of the genotypes presenting high stability. The smaller TOL values indicate the genotypes with low yield depression and hence more tolerant. However, good heat tolerance can also be reached by low yielding lines, but their value for breeding would be questionable. Hence, a scatterplot was devised to compare the GY under normal conditions and each of the heat indices (SSI and TOL). Five genotypes (four ICARDA lines, one Moroccan cultivar): Kunmiki, Berghouata1, Margherita2, IDON37-141, and Ourgh were found to have above average yield, low yield depression (low TOL values) and good heat tolerance (SSI < 1).

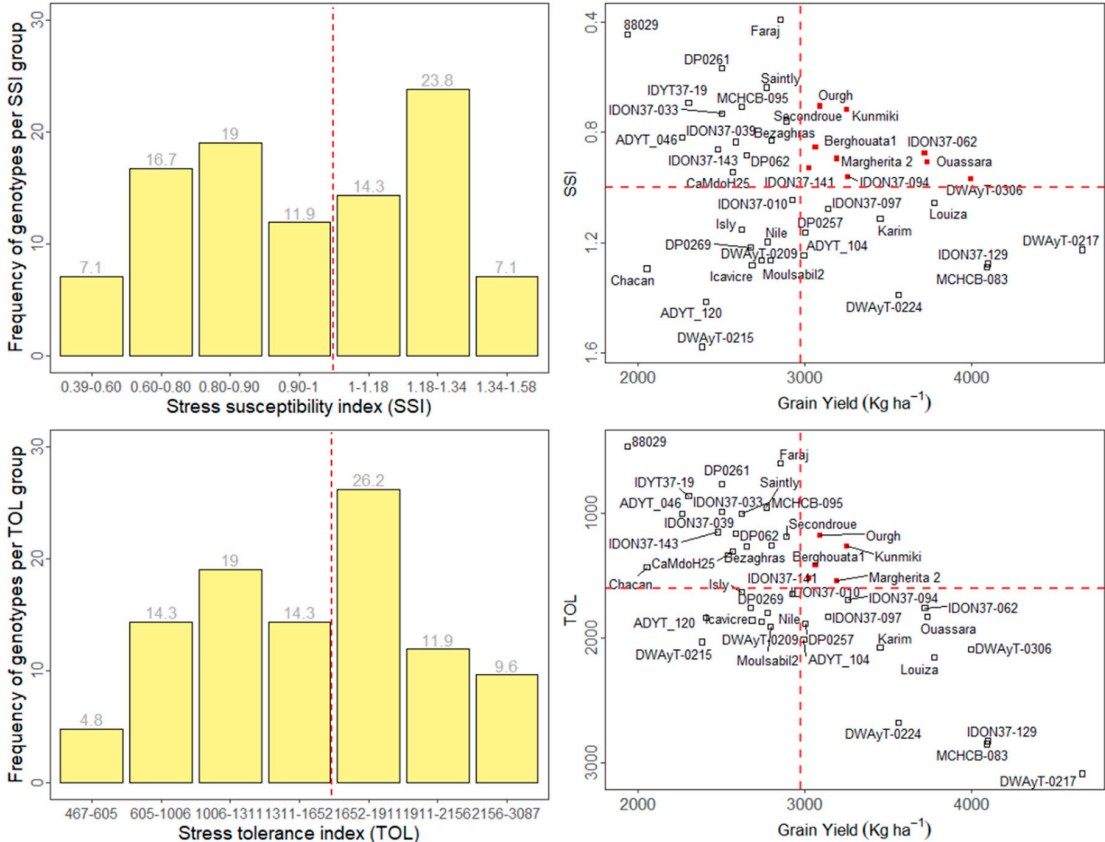

**Figure 4.** Two different stress tolerance indices SSI (stress susceptibility index) and TOL (tolerance index) of grain yield, comparing plastic tunnel-mediated heat stress with normal conditions for the 42 durum wheat genotypes. The bars plot shows the frequency distribution of SSI and TOL for the genotypes tested. The dashed red lines mark the separation between tolerant (left) and susceptible (right) genotypes. The scatter plot shows the yield performance of genotypes tested under normal conditions against each of SSI and TOL. The vertical dashed red lines indicate the average GY. The horizontal dashed red lines indicate the cut-off value for tolerant vs. susceptible genotypes for each index. Red dots indicate genotypes that were identified as superior by both bi-plots.

### 3.4. Markers Associated to Heat Stress Tolerance

A total of 204 MTAs were identified for four traits (GY, GNSpk, HI and Biom) under both stress and normal conditions and 49 MTAs were recorded for the two GY stress tolerance indices. Regression analysis and clustering based on local LD decay confirmed that these associations were distributed over 12 loci (Table 2 and Table S6). Chromosome 1A had the highest number of MTAs (27) while chromosome 4A had the lowest (6).

Under normal conditions, 56 MTAs were detected for three traits GY, GNSpk, and HI, with the third trait having the highest number of MTAs (48). No common region for these traits was identified under the non-stress environment. Under heat stress, a higher number of associations (148) were identified with trait variation ($r^2$) ranging from 0.25 to 0.36. The highest number of MTAs were detected for GNSpk distributed over 10 different loci, followed by HI on six loci. A common region for GY, GNSpk, HI, and Biom was identified under the heat condition on chromosome 6BS. Loci associated with both GNSpk and HI were detected on 1AL, 1BL, 2AL, 3AL, and 3BL. For heat tolerance indices (SSI-GY and TOL-GY), 49 MTAs were identified. The common loci associated with the two indices were on chromosomes 2AL, 5AL, and 5BL, while the loci on chromosomes 1AL and 6BS were identified only for TOL-GY and SSI-GY, respectively.

A comparison of the significant loci under each treatment and including the heat tolerance indices indicated a locus on chromosome 2AL, which was consistently identified for the indices, and both treatments for GNSpk and HI. Two loci on chromosomes 3AL and 3BL were associated with GNSpk and HI under both control and stress conditions, but were not associated with any of the indices. Three significant loci on chromosomes 1AL, 5BL, and 6BS were shared among heat stress treatment and stress tolerance indices, but not under normal conditions, making of these the most interesting genomic regions that specifically respond to heat stress. Overall, a total of 12 unique significant loci were identified (numbered QTL.ICD.Heat.01–QTL.ICD.Heat.12) and can be consulted in Table 2. Local LD decay was estimated for the 100 Mbp genomic region surrounding the peak marker. It varied between 31.7 and 108.7 Mbp, or a −38% to 112% variation compared to the average LD decay calculated for the whole panel (51.3 Mbp). This variation was accounted for to determine the correct physical size in each genomic region to assign multiple MTAs to the same QTL.

**Table 2.** Quantitative trait loci (QTLs) associated with multiple traits under plastic tunnel-mediated heat stress, normal conditions, and based on stress indices.

| Locus | Trait | Chr. [†] | Main Marker | Position [‡] (bp) | Local LD (Mbp) | Max LOD | Max $r^2$ | Heat Stress | Normal | Indices |
|---|---|---|---|---|---|---|---|---|---|---|
| QTL.ICD.Heat.01 | GNspk, HI, TOL-GY | 1AL | AX-94863732 | 570,040,339 | 31.7 | 3.38 | 0.27 | * | | * |
| QTL.ICD.Heat.02 | GNspk, HI | 1BL | AX-94447402 | 632,403,981 | 43.1 | 3.38 | 0.27 | * | | |
| QTL.ICD.Heat.03 | GNspk, HI, SSI-GY, TOL-GY | 2AL | AX-94538070 | 748,624,588 | 36.3 | 3.06 | 0.25 | * | * | * |
| QTL.ICD.Heat.04 | GY, HI | 2BS | AX-95193898 | 6,012,904 | 36.0 | 3.67 | 0.36 | | * | |
| QTL.ICD.Heat.05 | GNspk, HI | 3AL | AX-95632723 | 562,421,267 | 75.4 | 3.39 | 0.27 | * | * | |
| QTL.ICD.Heat.06 | GNspk, HI | 3BL | AX-95174625 | 788,551,042 | 85.4 | 3.38 | 0.27 | * | * | |
| QTL.ICD.Heat.07 | GNspk | 5AS | AX-95247611 | 27,923,949 | 108.7 | 3.38 | 0.27 | * | | |
| QTL.ICD.Heat.08 [§] | SSI-GY, TOL-GY | 5AS | AX-94631521 | 421,078,546 | 41.3 | 4.93 | 0.45 | | | * |
| QTL.ICD.Heat.09 [§] | GNspk, SSI-GY, TOL-GY | 5BS | AX-95182463 | 427,098,066 | 50.3 | 4.17 | 0.37 | * | | * |
| QTL.ICD.Heat.10 [§] | GNspk, HI, Biom, SSI-GY | 6BS | AX-94408589 | 157,777,006 | 56.0 | 3.20 | 0.36 | * | | * |
| QTL.ICD.Heat.11 | GNspk | 7AL | AX-95074729 | 660,833,752 | 153.6 | 3.60 | 0.29 | * | | |
| QTL.ICD.Heat.12 | GNspk, HI | 7AS | AX-94381852 | 16,943,364 | 44.8 | 3.42 | 0.37 | | * | |

[†] Chr.—Chromosome, based on alignment to durum wheat genome assembly [48].*—Significant QTL; [‡]—Based on alignment to durum wheat genome assembly [48]; [§]—These QTLs have been converted into KASP markers and validated; GNspk—Grain number per spike; HI—Harvest index; TOL-GY—Tolerance index for grain yield; SSI-GY—Stress susceptibility index for grain yield; GY—Grain yield; Biom—Biomass.

### 3.5. Effect of Different Allele Combination on Yield Performance

The loci identified on chromosomes 1AL, 5BL, and 6BS appeared as the most critical for heat tolerance and were then tested further. These regions were associated with the control of multiple traits under heat stress: GY, GNspk, HI, Biom and the two indices SSI-GY and TOL-GY. A set of 208 modern lines were investigated for haplotype diversity at these three loci. Five groups with different allelic combinations were identified (Figure 5). Their allelic effect on GY was then assessed when field tested under high temperatures along the Senegal River [35]. The haplotype class with positive alleles at all three loci had the highest GY average reaching 2381 kg ha$^{-1}$ with a maximum value of 3856 kg ha$^{-1}$. Genotypes of the haplotype classes with only two favorable alleles reached GY of 2199 and 2103 kg ha$^{-1}$, while lines that only carried one positive allele 2103 and 2023 kg ha$^{-1}$ (Figure 5). ANOVA confirmed that the haplotype group with all three positive alleles was significantly superior to the others.

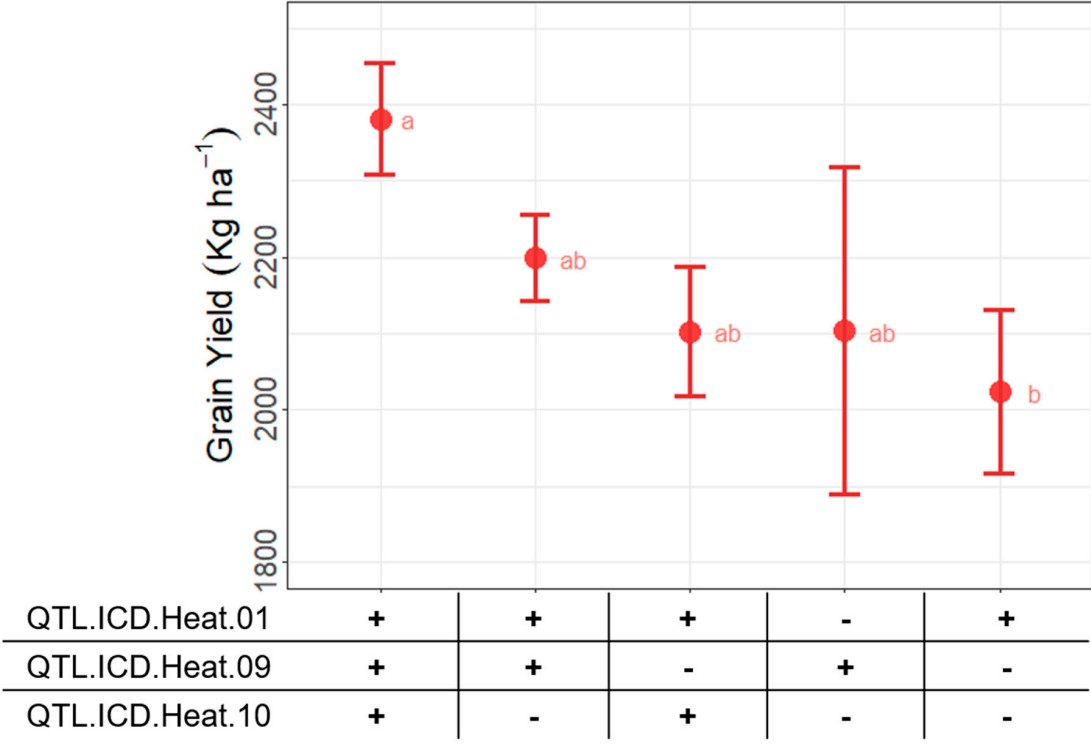

**Figure 5.** Effect of different allele combinations of the significant loci on yield performance of 208 accessions tested under heat stressed conditions along the Senegal River. The circle indicates the average of each class over 2 years, and the whiskers show the standard error of the mean. The accessions were divided into five clusters based on their haplotype for three major QTLs: "+" mark the positive and "-" the wild-type alleles. Letters (a, b, ab) indicate significant differences between the clusters.

### 3.6. Validation of Markers for Marker Assisted Selection

To effectively deploy in breeding the most interesting QTLs via MAS, it is first required a step of validation using more affordable marker methodologies and in different genetic backgrounds and environments. A total of 20 MTA sequences linked to important agronomical and spike fertility traits were submitted for KASP primers design. Among these, only 14 could be successfully designed, and 11 identified a polymorphism within the validation set. Four showed significant ($p < 0.05$) correlation to the test phenotype (Figure 6). Three QTLs were represented by these four markers, AX-95260810 and AX-94432276 tagged QTL.ICD.Heat.08 on chromosome 5AL, AX-95182463 underlines QTL.ICD.Heat.09 on chromosome 5BL, and AX-94408589 tags QTL.ICD.Heat.10 on chromosome 6BS. The latter two QTLs are among the three main effect regions identified in this study (Figure 5).

AX-95260810 reached 15% correlation to grain yield under heat, 74% accuracy, 43% sensitivity, and 100% specificity. Especially, its ability to identify 100% of non-heat tolerant entries is particularly remarkable. AX-95182463 and AX-94408589 also reached significant correlations of 14% and 32% for grain yield and biomass under severe heat, respectively, with sensitivities of 62% and 40%, accuracies of 30% and 65%, and specificities of 4% and 90%. Overall, AX-9526081 and AX-94408589 appeared as the most suitable for MAS application.

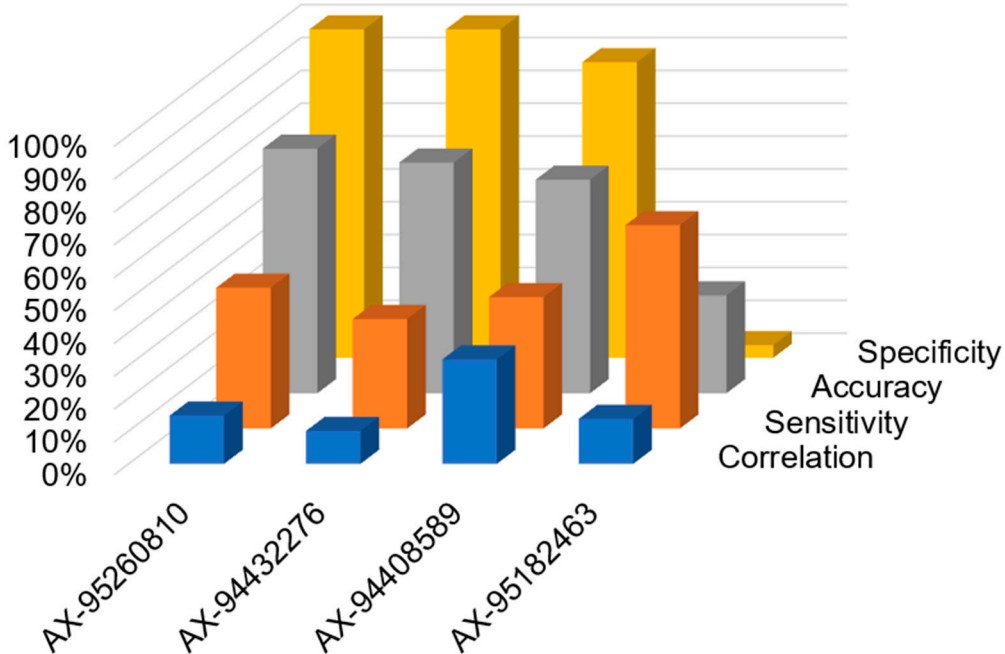

**Figure 6.** Kompetitive Allele Specific PCR (KASP) markers validation on an independent set of 94 elite lines of ICARDA tested under severe heat for grain yield and biomass. Correlation was measured between the BLUE for grain yield recorded along the Senegal River and the haplotype score. Accuracy, sensitivity, and specificity where determined using only the top 20 and worst 20 lines. AX-95260810 and AX-94432276 tag QTL.ICD.Heat.08, AX-95182463 tags QTL.ICD.Heat.09, AX-94408589 tags QTL.ICD.Heat.10.

## 4. Discussion

*4.1. Evaluation of the Phenotypic Performance of Yield and Yield Components under Normal and Heat Stress Conditions*

Several studies reported that wheat plants are very sensitive to elevated temperatures during flowering and grain filling phases [9,55,56], due to a reduction in seed development and fertility [56–58]. This study evaluated a set of durum wheat genotypes derived from a global collection for GY and yield components under heat and normal conditions. The genetic and phenotypic diversity shown by this set together with its relatively similar flowering time, promote it as an ideal panel to test heat tolerance. Further, the plastic tunnel method deployed here allowed to increase the temperatures well above 21 °C, the value that defines the absence of the stress [9]. A similar methodology was also successfully deployed by Corbellini et al. [54] to study the effect of heat shock proteins on technological quality characteristics. Compared to timely vs. delayed sowing experiments to simulate heat stress, the use of the plastic tunnel method avoids incurring false discovery due to changes in the phenological behavior of plants.

In the present study, a short and severe episode of heat stress was applied from the beginning of heading to the early dough stage, and resulted in 54% reduction in grain yield. This was in agreement with the study conducted by Ugarte et al. [59] that found a reduction of up to 52% when thermal

treatment was applied via transparent chambers. Interestingly, our stress treatment caused an average temperature increase of 10 °C, which caused an average GY reduction of 5.4% for each 1 °C raise. This value is well within the 4.1% to 6.4% interval suggested by Liu et al. [10] for 1 °C raise in temperatures. GNSpk was the most affected trait (−61%) with the highest positive correlation to GY. This is in good agreement with previous studies that have shown that seed setting is the most sensitive parameter to heat stress, with a noticeable influence on yield [28,60–62]. Still, its non-linear relationship to yield confirms the complexity of the trait. Biom and HI were also found to have an influence on yield [63,64] with different relationships based on the occurrence of the stress. The presence of dissimilarities of the associations between the two treatments indicates clearly that there is a trade-off among the yield components as previously reported by Sukumaran et al. [65] for grain weight and grain number. Variation of one of the yield components affect the others positively or negatively. Compared to the simple regression, the additive model allowed to reveal the complexity of the relationship between GY and yield related traits.

The stress index SSI was developed by Fisher and Maurer [43] and modified by Nachit and Ouassou [44] as a useful indicator and a good parameter for selection. It measures the severity of the heat stress [66,67] and was also used in earlier studies in wheat to seek heat tolerant genotypes [23,68,69]. The TOL index is instead useful for selecting against yield depression, and it was used in several studies for heat or drought tolerance in wheat [27,44,67,70]. Improving heat tolerance should not be based on the use of these criterions alone as was suggested by Clarke et al. [71]. It is important to select simultaneously for good yield performance coupled with good adaptability (SSI < 1) and stability (low TOL) [44]. In that sense, the accessions Kunmiki, Berghouata1, Margherita2, and IDON37-141 originated from ICARDA durum wheat program, and Ourgh, a Moroccan cultivar, have been identified as high yielding genotypes that also show good heat stress tolerance based on the two indices.

## 4.2. Dissection of Heat-Specific QTLs Associated with Yield-Related Traits and Stress Tolerance Indices

The significant correlation identified between yield and its components were not linear in nature, and tend to change their mode of action based on the occurrence of the stress. Therefore, several physiological processes are simultaneously involved in protecting the wheat plant from the heat stress [72], and there is value in dissecting it into its genetic components. In this study GWAS was used to identify the genetic regions controlling the response of the various traits. To prevent the confounding effect that phenology-related loci might have [73], MTAs were identified for DTH and removed from downstream analysis. Additionally, flowering time was used as covariate in all analyses for the other traits. Very few MTAs for DTH were observed either in normal or stressed conditions due to the synchronized flowering of the entries used in this study. This indicated the absence of confounding effects between the two trials. i.e., almost all the accessions were exposed to the same conditions in each developmental phase [74] before imposing the stress.

Out of 12 QTLs identified, three occurred only when the heat stress was imposed, including indices. These three main genomic regions occurred on chromosomes 1AL, 5BL, and 6BS, and were considered as QTLs controlling heat tolerance. These three loci were confirmed by mean of haplotype analysis on a larger panel of modern lines (208 entries) field tested under severe heat along the Senegal River valley [35], to confirm that the presence of the positive alleles at all three loci provided a significant GY advantage of +182 kg ha$^{-1}$ (+8%). The QTL on the long arm of chromosome 1A controlled GNSpk, HI, and TOL-GY, and it explained up to 27% of the phenotypic variation. In a study with double haploid population of bread wheat, Heidari et al. [75] identified a major QTL on the same chromosome (1A), influencing grain number per spike, grain weight per spike, and spikes/m$^2$. However, their phenotypic assessment was not performed under heat stress, the marker systems used was different compared to our study and the locus was identified in the short arm of chromosome 1A. Therefore, it is quite difficult to align the results from that study to the current one. Another study had previously reported many MTAs on chromosome 1A detected for yield components under heat stress, but all were found to have a pleiotropic relationship with days to heading and were also located on the short arm of 1A [26],

instead of 1AL found here. A heat-specific QTL was also detected on the same chromosome in the short arm for spikelet compactness and leaf rolling in bread wheat [76]. An earlier study identified a QTL on 1AS for yield but associated with different stress conditions [77]. To the best of our knowledge, this is the first time that this region on 1AL is presented as associated to GNSpk, HI, and TOL-GY in durum wheat under heat stress conditions. The second major QTL region was detected on the long arm of chromosome 5B and found to be associated with GNSpk and the two indices SSI-GY and TOL-GY, contributing to 37% of the phenotypic variation. A region in the short arm of the same chromosome has been previously reported to be associated with grain number per square meter in bread wheat [76], and controlling thousand grain weight in durum wheat [27] under combined drought and heat stress. Shirdelmoghanloo et al. [25] and Acuna-Galindo et al. [78] reported loci for grain weight and other important traits on chromosome 5B under heat and non-heat conditions in hexaploid wheat. On the other hand, the same chromosome has been previously suggested to carry heat-specific QTLs for yield per se in bread wheat [26]. Sukumaran et al. [27] identified markers for heat susceptibility (HSI or SSI) and tolerance (TOL) indices for yield and grain number per square meter on the short arm of the chromosome 5B. Mason et al. [64] also detected QTL for HSI for kernel number on 5BL in bread wheat. The genomic region identified in this study on 5BL is likely to be a new QTL since no information has been reported earlier for this locus associated to GNSpk, SSI-GY, and TOL-GY in durum wheat and specific to heat stress, but we cannot exclude that it overlaps with previously reported QTLs. A third heat-responsive locus was identified on the short arm of chromosome 6B related to GY, SSI-GY, GNspk, HI, and Biom accounting for 36% of the phenotypic variance. An earlier study on bread wheat identified a locus on chromosome 6BS underpinning chlorophyll loss rates and heat susceptibility index for grain weight and chlorophyll loss rates under heat-stress conditions [25]. Under post-anthesis high temperatures stress, Vijayalakshmi et al. [20] reported a QTL on the short arm of chromosome 6B for senescence related traits in hexaploid wheat. McIntyre et al. [79] and Pinto et al. [21] reported QTLs on chromosome 6BL that were associated with many important traits (grain number per square meter and grain yield and water-soluble carbohydrate content) related to drought and heat tolerance. Ogbonnaya et al. [26] found a locus on the short arm of chromosome 6B for grain yield under heat stress in bread wheat. These previously reported QTLs in 6B could overlap with the one identified in this study, but they were either identified not in association with heat tolerance or detected in hexaploidy wheat. Therefore, this region is also assumed to have been reported for the first time here in relationship to heat tolerance for durum wheat. This locus affects multiple traits (GY, GNspk, HI, Biom, and two heat susceptibility indexes) and hence it is of good importance for deployment in breeding. The principal breeding objective is to develop varieties with high grain yield and stability when exposed to different stresses. However, grain yield is a complex trait controlled by many genes and strongly influenced by the environment [80–86]. Therefore, a good understanding of traits and underlying loci associated with tolerance to elevated temperatures is of a great importance for breeding new heat tolerant cultivars [87].

*4.3. Pyramiding Heat-Tolerant QTLS via MAS*

Three loci on chromosomes 1AL, 5BS, and 6BS showed an additive nature by means of haplotype analysis (Figure 5), revealing that only the combination of all three positive alleles generated a true yield advantage. Among the most heat tolerant elite lines identified here 'Kunmiki', 'Berghouata1', and 'Ourgh' confirmed to harbor the positive alleles for all three loci. This prompts their use in crossing schemes to pyramid the positive alleles, as well as the deployment of simple marker system to conduct MAS.

Axiom to KASP marker conversion and validation was attempted for 20 MTAs. Eleven KASP markers generated polymorphic haplotypes in an independent set of ICARDA elite lines. Four revealed a significant ($p < 0.05$) correlation to GY and biomass assessed under severe heat along the Senegal River Valley (Figure 6). In particular, AX-95182463 tags QTL.ICD.Heat.09 located on chromosome 5B and it revealed good correlation and sensitivity, but lacks in accuracy and specificity, and it is hence

protected from Type II errors, but prone to Type I, with several elite lines wrongly identified as carrying the positive alleles. AX-95260810 tags QTL.ICD.Heat.08, linked to the two stress tolerance indices for GY (SSI-GY and TOL-GY) located on chromosome 5A. AX-94408589 tags QTL.ICD.Heat.10 located on chromosome 6B, and associated to several traits GNspk, HI, Biom, SSI-GY. In these two cases, the KASP markers explained 15% and 33% of the phenotypic variation of an independent validation set, with 100% and 90% specificity, and 74% and 65% accuracy, but medium sensitivity (43% and 40%). As such, these markers are protected against Type I errors (no false positive), but prone to Type II errors, with several elite lines identified as not carrying the positive allele while instead being tolerant to heat. Hence, while all converted KASP markers are prone to different types of errors, these three markers can be considered as validated and ready to be deployed in breeding. The combination of the three might represent a more stringent approach to protect against both types of errors. An additional nine QTLs were identified in this study, and their KASP conversion and validation are still ongoing and will require better targeted efforts to be achieved.

## 5. Conclusions

Heat stress causes a complex cascade of negative effects on the wheat plant, resulting in drastic reductions in grain yield. The deployment of heat tolerant varieties that will benefit greatly farmers requires first to enhance our understanding of this mechanism and loci governing it. Our study combined a discovery phase with a core set tested over two field seasons in Morocco under artificial heat-treatment with plastic tunnels, followed by a different confirmation set of germplasm grown for two seasons in Kaedi, Mauritania under severe natural heat, and completed with one final validation set tested one season in Kaedi. Our results confirmed that spike fertility (GNSpk) and maintenance of green leaves (Biom) are the most critical traits to drive tolerance to this stress, and hence should be the primary targets of durum wheat breeders. Further, the deployment of plastic tunnels proved to be a strategic methodology to study this stress and reveal its mechanisms without affecting the phenology of the plant. In addition, 12 loci were identified as responsible for controlling the main heat tolerance traits. Among these, three were activated only when the stress occurred and hence represent ideal targets for breeding. Two of these were validated into a KASP marker and are now ready for deployment via MAS, especially if associated with a third, also validated, KASP. Finally, three ICARDA elite lines and one Moroccan cultivar were confirmed as tolerant to heat, with high grain yield, and carrying positive alleles for three main QTLs. These are freely available and should be incorporated as crossing parents by other breeding programs. Altogether, this study has confirmed the key traits for heat tolerance as well as a new methodology to study it in durum wheat, it has revealed the main loci controlling these traits and proceeded to validate three of them for MAS, and it has also provided freely available elite lines to breed new cultivars better adapted to the stress.

**Supplementary Materials:** Table S1: List of durum wheat genotypes evaluated under plastic tunnel-mediated heat stress in the present study, Table S2: Markers associated with days to heading (DTH) under heat stress and normal conditions, Table S3: Sequence information of the KASP markers, Table S4: Pearson correlation matrix between all the measured traits under heat conditions (upper part) and normal (lower part) conditions. GY—Grain yield; Biom—Biomass; HI—Harvest index; Spkm$^2$—Spikes per square meter: GNspk—Grain number per spike; TKW—Thousand kernel weight; DTH—Days to heading. *, ** Significant at the 0.05 and 0.01 probability levels, respectively, Table S5: Correlation (r), linear regression estimated via ordinary least squares (OLS) and flexible regression estimated via regression additive model. (**a**) Under heat stress. (**b**) Under normal conditions, Table S6: Regression matrix between the haplotype of the peak markers for the 13 identified QTLs. *, significant loci similarity at $p < 0.05$ consistent with homeologous relationship; **, significant loci identity ($p < 0.01$) consistent with wrongly assigned genomic position, Figure S1: Plots of the additive regression model showing GNspk, biom, TKW, spkm$^2$ and HI as the spline function of the target trait grain yield (GY). (**a**) Under heat stress. (**b**) Under normal conditions

**Author Contributions:** F.M.B., M.N. and K.E.H. conceived and designed the study. K.E.H. and F.M.B. performed the field experiment. A.T.S. performed the field experiment in the Senegal river. A.A. contributed in the genotyping. K.E.H. and F.M.B. analyzed the data. K.E.H. Wrote the original draft. K.E.H., B.B., A.F.M., A.A., M.N., and F.M.B. wrote or reviewed the manuscript. All authors read and approved the final manuscript.

**Funding:** This study was funded by the Australian Grains Research and Development Corporation (GRDC) project ICA00012: Focused improvement of ICARDA/Australia durum germplasm for abiotic tolerance, while the field work along the Senegal River was funded by the Swedish Research Council (Vetenskapsradet) U-Forsk2013 project 2013-6500, "Deployment of molecular durum breeding to the Senegal Basin: capacity building to face global warming" and U-Forsk2018 project 2017-05522, "Genomic prediction to deliver heat tolerant wheat to the Senegal River basin: phase II." The marker conversion work was covered by the International Treaty on Plant Genetic Resources for Food and Agriculture 2014-2015-2B-PR-02-Jordan: "An Integrated Approach to Identify and Characterize Climate Resilient Wheat for the West Asia and North Africa."

**Acknowledgments:** The authors wish to acknowledge the technical assistance provided by A. Rached and all ICARDA durum wheat program staff in handling field activities.

**Conflicts of Interest:** The authors declare no conflict of interest.

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
