# Peer review of "Loci Controlling Adaptation to Heat Stress Occurring at the Reproductive Stage in Durum Wheat"

_agronomy, doi:10.3390/agronomy9080414_

Reviewer 1 Report

Overall this study was well designed and presented. There are some minor changes that could be made:

Abstract:

Line 19 should be “elite lines” not elites.

Line 21: should state significant instead of critical

Intro

Line 42 replace good with better

Materials and methods

Line 72: should read “composed of 94 of ICARDA’s elite lines

Authors should avoid using the term “elites” and use elite lines instead.

Results

Figures 2, 3 and 4 are low quality and blurry.

Table 2: The R2 values are very high and it appears that some markers on different chromosomes may be duplicates – possible due to errors in the genome sequence. For example, Markers on 3A, 3B and 5A have nearly identical R2 and LOD values. The same is true for 1A and 1B. Possibly due to homologs, but some explanation or clarification is needed.

Author Response

Please kindly find point by point response attached

Reviewer 2 Report

The manuscript by El Hassouni et al describes a series of field experiments carried out with different subsets of durum wheat lines in order to identify 3 KASP markers that breeders could use to do marker-assisted selection of lines tolerant to heat stress. The paper is well-written and readers will have little problem following the flow.

I am not a breeder myself so I cannot fairly critizise some aspects of the work, so I will split my comments in two groups: breeding-related (where I am an informed reader) and genome-related (really my area).

# Breeding-related comments

1) The authors say that "heat stress tolerance QTLs are not generally used in MAS in wheat". This is a bold statement and perhaps should be modified to add citations of other papers discussing the same topic, such as:

https://www.researchgate.net/publication/279757599_Breeding_for_heat_tolerance_in_Wheat
https://onlinelibrary.wiley.com/doi/full/10.1111/pbr.12217
https://link.springer.com/article/10.1007/s00497-016-0275-9
https://www.sciencedirect.com/science/article/pii/S221451411730096X

2) The authors should discuss the limitations of using only two years of field data on one location. That would not be enough to register a new v
ariety in some Europen countries.

3) I suggest the authors cite the papers by the group of Slafer to compare their plastic tunnels to theirs.

# Genome-related comments

4) Is 98% identity enough to map subgenome-specific sequences, did the authors look for possible ambiguities?

5) Assuming that LD decay is constant across the genome, with a fixed window size of 51.3Mb might be a little bit over-stretching, as we know that LD is actually higher usually as you get closer to the centromeres. I suggest the authors define moving windows of variable size to account for
 this.

6) With respect to the KASP markers I have several comments:

6.1) Can we also see specificty & sensitivity for the two separate years? Was this all done in the same location? See comment 2.

6.2) Where is the sequence of the KASP markers for others to use? Is it enough to have the Ax accession for that? See for instance an example in Enseml Plants:
http://plants.ensembl.org/Triticum_aestivum/Marker/Details?db=core;g=TraesCS4A02G446800;m=Cadenza1704.chr4A.714191310;r=4A:714191210-714195148;t=TraesCS4A02G446800.1;v=Cadenza1704.chr4A.714191310;vdb=variation;vf=21748957
-- INSERTAR --                                                                   

Author Response

Please find point-by-point response in the attached document
